# Small target detection algorithm based on multi-branch stacking and new sampling transition module

**Qingyao Lin**☯, **Rugang Wang**☯*, **Yuanyuan Wang, Feng Zhou**

School of Information Technology, Yancheng Institute of Technology, Yancheng, China

☯ These authors contributed equally to this work.

* wrg3506@ycit.edu.cn

**Data Availability Statement:** All relevant data are within the manuscript and its Supporting information files.

**Funding:** This work was supported by the Jiangsu Graduate Practical Innovation Project (No:

## Abstract

Aiming at the problem that the SSD algorithm does not fully extract the feature information contained in each feature layer, as well as the feature information is easily lost during the sampling process, which makes the feature expression ineffective and leads to insufficient performance in small target detection. In this paper, AMT-SSD is proposed, a small target detection algorithm that incorporates the multi-branch stacking and new sampling transition module of the attention mechanism. In this algorithm, the composite attention mechanism is utilized to improve the correlation of features of the samples to be detected in terms of spatial and channels, and the efficiency of the algorithm; secondly, multi-branch stacking module is used to extract multi-size features for each feature layer, and different sizes of convolution kernels are utilized in parallel to fully extract their features and improve the expression of features; meanwhile, during the sampling process, the problem of missing features is solved by applying inverse subpixel convolution in the new sampling transition module. Experimentally, the AMT-SSD algorithm achieves 84.6% and 53.4% mAP metrics on the PASCAL VOC dataset and MS COCO dataset, respectively. This indicates that the AMT-SSD algorithm can effectively extract feature information that is beneficial to detection samples, and also performs well in reducing feature loss, which is effective for the algorithm to improve the algorithm on small targets.

## Introduction

There are many important downstream branch tasks in the field of computer vision, and target detection, as an important one, has achieved remarkable results in various fields. Zhang [1] et al. proposed an ALS-Performance Releaser (PRSAL) with the learning function of intelligent anchors for anchor frame learning strategies. It utilizes anchor classification ability as an equivalent indicator of anchor box regression ability to screen anchors with high detection potential in a more rational way. Chakraborty R [2] et al. proposed an improved version of Fractional Order Darwinian PSO (IFODPSO) for segmenting histogram based 3D color images on multiple levels of Berkeley Segmentation Data Set (BSDS500). This scheme

SJCX22_1685), the Major Project of Natural Science Research of Jiangsu Province Colleges and Universities (No: 19KJA110002), the Natural Science Foundation of China under Grant (No. 61673108), the Natural Science Research Project of Jiangsu University (NO. 18KJD510010). The sponsors are Qingyao Lin, Rugang Wang and Feng Zhou. And Lin served primarily in the implementation of the study and writing of the manuscript, and Wang and Zhou served primarily in manuscript layout and content direction.

**Competing interests:** The authors have declared that no competing interests exist.

overcomes the complete dependence on score coefficients when dealing with multilevel problems with datasets, which provides new ideas for studying computer vision segmentation tasks. Zhou M [3] et al. demonstrated a robust framework for needle detection and localization in robot-assisted subretinal injections using a cross-cutting study between deep learning and medical contexts. The best performing network successfully detected and localized all needles in the dataset with an IoU value of 0.55 when evaluated on live pig eyes. Manjari K [4] et al. developed a novel lightweighting model with ResNet50 and MobileNetV2 as the backbone, which improves the efficiency of scene text detection and reduces the resource cost. It provides ideas for our subsequent research on network lightweighting.

With the rapid development of convolutional neural networks, target detection algorithms have also made continuous progress, and a variety of schemes have emerged to solve various types of problems in the target detection task from different perspectives [5–17]. Although the concept of multi-scale detection is proposed in the traditional SSD algorithm, its network does not fully consider the correlation between different convolution layers. The transfer of feature information between models is also unidirectional. With the convolution function layer As the number of layers increases, the semantic information will also be lost, resulting in the feature information between different feature layers not being fully utilized, thus affecting the detection effect of small targets. In response to the above problems, experts and scholars have conducted a lot of research work. Li et al. and Liu et al. proposed the FSSD algorithm and RFB-Net algorithm respectively by utilizing the characteristics of SSD algorithm that can extract feature maps at different scales for its multi-scale fusion. The FSSD algorithm combines three shallow features in SSD. The layers are fused once and a feature pyramid is generated, which improves the detection effect of the algorithm on small targets. However, the FSSD algorithm does not fully utilize the feature information of each layer during the feature fusion process, and makes less use of the semantic information of the features of each layer. RFB-Net, on the other hand, improves the detection effect by borrowing from the Inception structure and expanding the receptive field of the algorithm, but it does not consider the role of information exchange between different feature layers [18, 19]. Li Y T et al used MobileNet combined with the SSD algorithm to optimize and adjust the network structure and parameters, compressing the model size while maintaining accuracy, but did not take into account the correlation between different feature layers [20]. Li H T and others selected feature layers of different scales to form a feature pyramid and fused scale-invariant convolutional layers to generate a set of feature layer mappings that enhance feature information, thereby improving the detection effect of the algorithm on small targets, but there are still The misdetection of similar objects requires further optimization and improvement [21]. WANG H and others proposed an optimized deformable region convolution model for the convolutional functional layer. This model uses the online hard example mining algorithm (OHEM) and the soft non-maximum suppression algorithm (S-NMS) to Improve detection results [22]. Lim, Zhao, Wang, Zhang et al. weighted the channels and space of the feature graph by adding an attention mechanism to the network model to enhance the algorithm's ability to perceive semantic features, reduce background interference, and improve detection efficiency. The difference lies in the use of different attention models and different feature fusion strategies [23–26].

Since the above target detection schemes based on the SSD algorithm do not fully explore the feature information contained in each feature layer and express it, as well as ignore the correlation between different feature layers. In this paper, based on the SSD algorithm, a small target detection algorithm called AMT-SSD that incorporates the attention mechanism of multi-branch stacking and a new sampling transition module is proposed. Our contributions are as follows:

1. The composite attention module is applied in the algorithm to filter the background and unnecessary redundant information, and the algorithm's focus on the main target information is increased while reducing the algorithm's intake of useless information, which also improves the processing efficiency of the detection task.

2. Then the multi-branch stacking module is used to process the extracted feature layers. Convolutional feature layers of different scales are used to perform adequate feature extraction for each feature layer, which achieves the goal of avoiding the loss of semantic information.

3. the new sampling transition module is used to realize the feature fusion between different scale feature layers. Under the premise of ensuring that enough feature information is extracted, the information exchange between the cross-feature layers is accomplished, so as to enhance the detection effect of the algorithm on small targets.

## Related work

### SSD algorithms

The SSD (Single Shot MultiBox Detector) algorithm is a single-stage target detection algorithm that uses VGG16 as its backbone network. The specific structure is shown in Fig 1. The use of different scale feature layers for detection and classification was first proposed in the SSD algorithm, and this pioneering design gives the SSD algorithm an advantage over other single-stage algorithms when dealing with small targets. In contrast to the two-stage algorithm, the

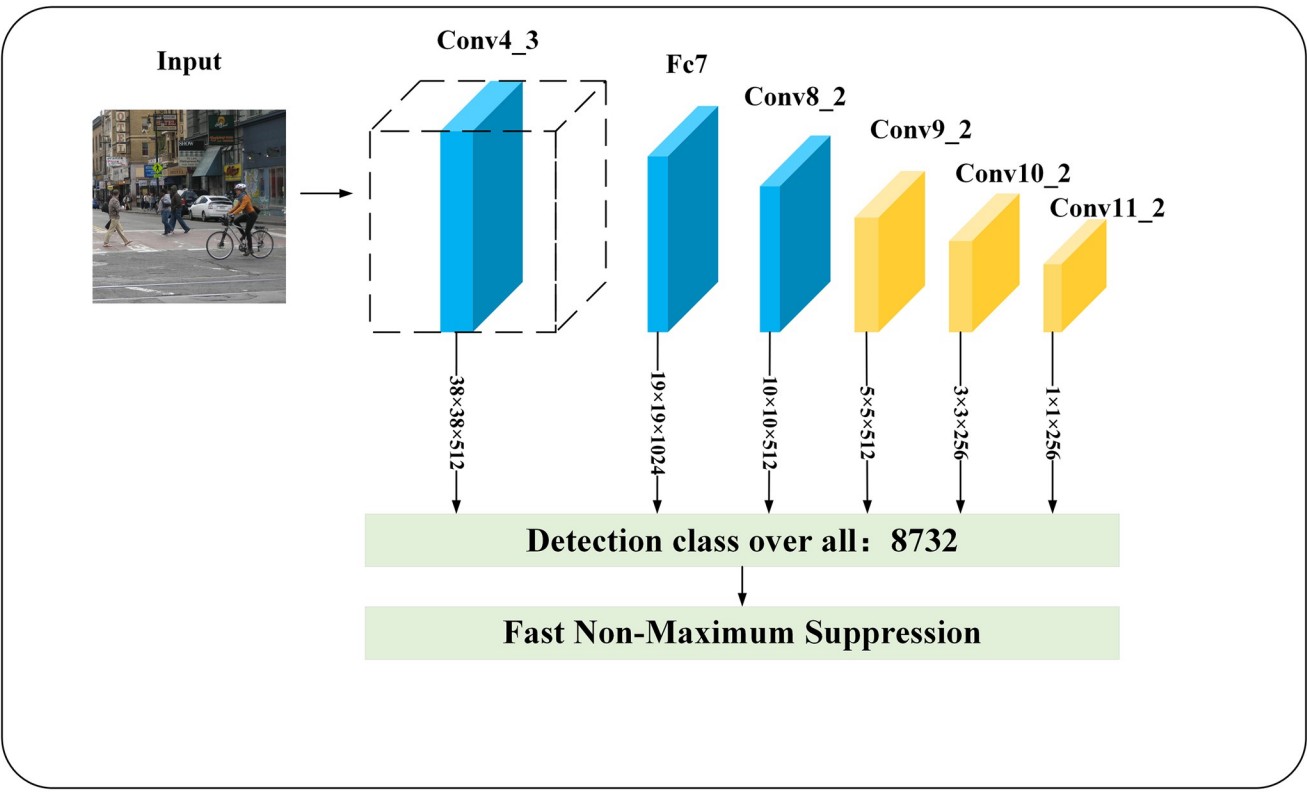

**Fig 1. SSD algorithm framework structure diagram.**

SSD algorithm provides direct prediction of the target without using RPN (Region Proposal Network) in detecting the target samples. Extracting multi-scale feature layers is designed to make the SSD algorithm perform well in terms of both detection accuracy and detection efficiency. However, in the SSD network, the feature layers are fragmented and the feature correlation between them is not fully utilized, which results in poor results when dealing with small targets and occluded targets. And Fig 1 is SSD algorithm framework structure diagram.

## Feature fusion solution

Feature fusion is the fusion processing of multiple feature layers at different scales to promote information fusion between different feature layers, thereby achieving the purpose of improving the performance of target detection tasks. In target detection tasks, the feature layers proposed by the backbone network often have obvious semantic differences due to lack of fusion. The shallow feature layers are rich in details that facilitate small target detection and localization, and the deep feature layers have more semantic information that is favorable to detecting medium and large targets. Therefore, how to make good use of the information in different feature layers has always been the research direction of scholars. With the continuous development of deep learning frameworks and hardware platforms, the design of feature pyramids is constantly being optimized and improved.

A bottom-up fusion path is added in PANet which based on the feature pyramid. While the expression of contextual information in the feature layer has been enhanced to some extent, the computational complexity of the network has also been significantly increased [27]. A method capable of adaptively adjusting the spatial weights of feature layers at different scales during the fusion process was proposed in ASFF. It not only reduces a certain amount of computation, but also allows for more efficient extraction of features [28]. The approach of first using different sampling methods to scale the feature layer uniformly, and then using a non-local network to capture global features to obtain balanced semantic information after integration has been applied in structures such as the BFP [29, 30]. Bidirectional information conversion between deep and shallow feature layers is realized in TPNet, which effectively improves the detection accuracy of the network for targets of different sizes [31]. Since traditional FPN cannot well solve the problem of fusing feature information at different scales, the proposal of BiFPN has opened up a new idea for researchers. The connectivity approach for transmitting contextual information is introduced in BiFPN while retaining the advantages of FPN and enhancing the fusion effect by adding different weights [32]. No matter which feature fusion model is used, it is a continuous attempt to find the best way to balance detection speed and detection accuracy.

## Analysis of the AMT-SSD algorithm model

Thanks to the design of outputting six different scale feature layers, the SSD algorithm has a natural advantage over other algorithms in detecting different target samples of large, medium and small sizes. However, the SSD algorithm does not adequately extract the feature information of each different feature layer, which leads to an unsatisfactory representation of the feature information in the feature layers. Moreover, each feature layer is independent of each other and does not utilize its correlation, which also makes the performance of small target detection insufficient. To address this problem, AMT-SSD is proposed in this paper, a small target detection algorithm that incorporates the multi-branch stacking and new sampling transition module of the attention mechanism. The specific structure is shown in Fig 2. The entire model structure consists of three parts: Composite Attention Module (CAM), Multi-branch stacking module (MSB), and Sample Transition Block (STB). Composite attention module is

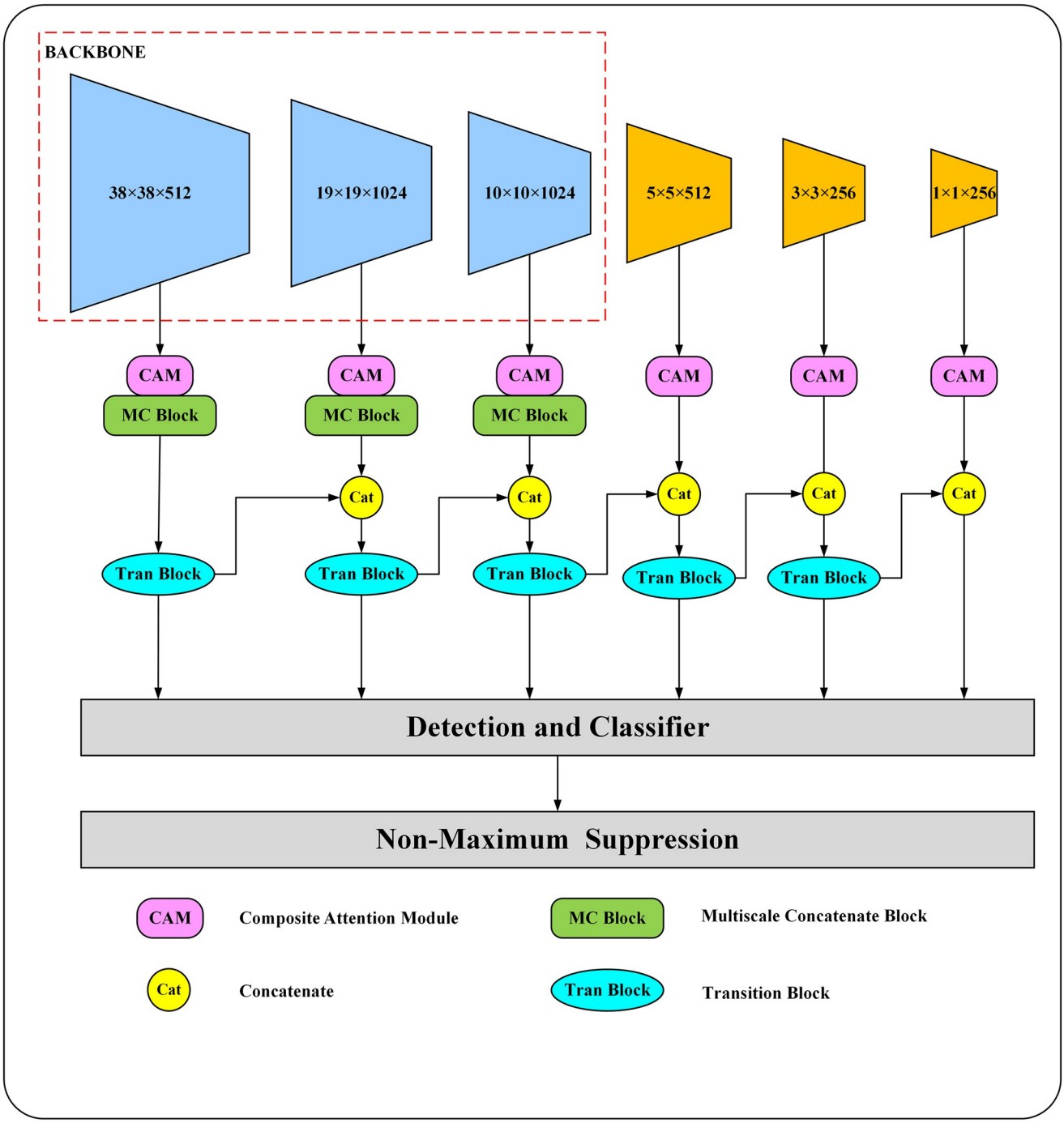

**Fig 2. AMT-SSD algorithm overall structure diagram.**

used to filter the useless in different feature layers and emphasize the feature information that is beneficial for performing target detection. Reducing the interference of useless information on the algorithm's reasoning process ensures the robustness of the algorithm; Multi-branch stacking module is used to process the shallow feature layer and utilize the convolutional functional layers with different scales from small to large for feature extraction to improve the

expression of feature information; In the sampling transition module, the down-sampling of large-size feature maps and their fusion with small-size feature maps are realized to promote cross-channel exchange of feature information. The information loss caused by convolution is compensated, and the feature layers are also enhanced to further improve the algorithm's detection performance on target samples. And Fig 2 is AMT-SSD algorithm overall structure diagram.

## Composite attention module

In order to alleviate the differences between channel information and spatial information due to different feature maps, reduce the redundancy of background interference and useless information, and improve the robustness of the algorithm model. In this paper, a composite attention module is designed to reduce the negative impact on the algorithm model due to the above reasons, and the specific structure is shown in Fig 3. The same feature layer is processed

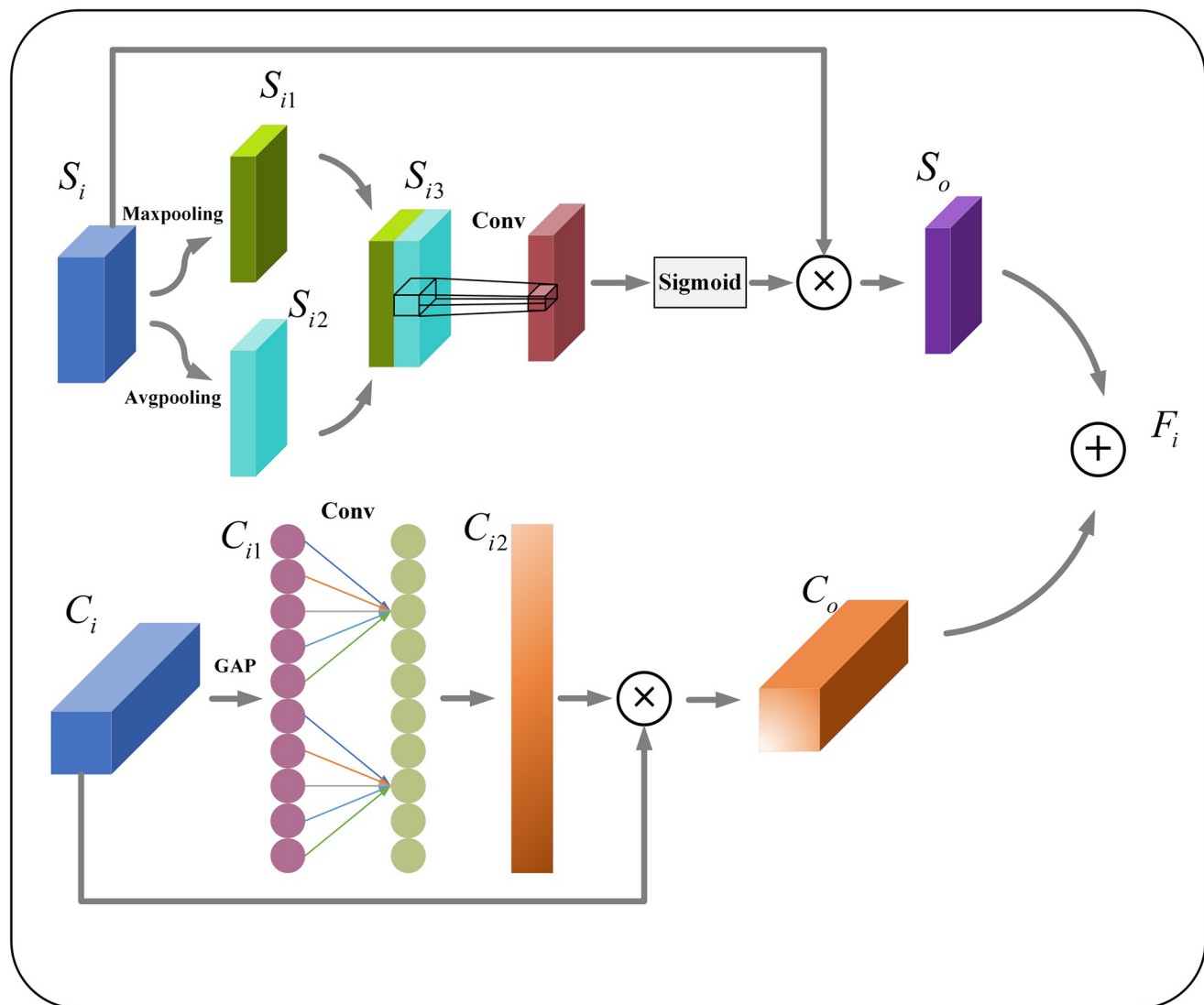

**Fig 3. Composite attention module structure diagram.**

in parallel for spatial attention and channel attention to extract spatial location information and semantic information related to the target detection samples, respectively. The extracted feature layer is fed into the spatial attention section to produce a spatial attention feature map by means of positional relationships within the feature map. The extracted feature layer is fed into the spatial attention section to produce a spatial attention feature map by means of positional relationships within the feature map. Maximum pooling and average pooling are performed on the feature map $S_i$ to obtain the feature maps $S_{i1}$ and $S_{i2}$; Then, the spatial attention feature map $S_o$ is obtained through convolution and Sigmoid activation function processing and multiplication with the original feature map $S_i$; $C_{i1}$ is obtained by feeding the same feature layer into the global average pooling. After that, 1D convolution is directly applied instead of fully connected layers to obtain better cross-channel information acquisition with less overhead. And the multiplication operation is performed with the feature layer $C_i$ to obtain the channel-attention feature map $C_o$. The spatial attention feature map $S_o$ is added with the channel attention feature map $C_o$ to obtain the final feature map $F_i$ with spatial location and channel semantic correlation. The processing of the composite attention module can be represented as follows:

$$
\begin{cases}
S_{i1} = \mathrm{Maxp}\,(S_i) \\
S_{i2} = \mathrm{Avgp}\,(S_i) \\
S_{i3} = S_{i1} \oplus S_{i2} \\
S_o = \mathrm{Sig}\,(\mathrm{Conv}\,(S_{i3})) \otimes S_i \\
C_{i1} = GAP(C_i) \\
C_{i2} = \mathrm{Sig}\,(\mathrm{Conv}\,(C_{i1})) \\
C_o = C_{i2} \otimes C_i \\
F_i = S_o \oplus C_o
\end{cases}
\tag{1}
$$

In the above formula, $Maxp()$ represents the maximum pooling operation, $Avgp()$ represents average pooling operation, $\oplus$ represents the addition operation of different feature layers, $Conv()$ represents the convolution operation, $GAP()$ represents the global average pooling operation, $Sig()$ represents the Sigmoid activation function operation, $\otimes$ represents the multiplication operation between different feature layers. And Fig 3 is Composite attention module structure diagram.

## Multi-branch stacking module

The SSD algorithm proposes to utilize feature layers of different scales to detect targets of different sizes. However, the information of different feature layers is not fully utilized, so that a large amount of valuable feature information is subsequently lost in the sampling process. To address this problem, a multi-branch stacking module is proposed and applied to different feature layers in this paper. In this structure different sized convolutional kernels are utilized in parallel to fully extract their features and improve the feature representation. The specific structure of the multi-branch stacking module is shown in Fig 4. The final stack fusion implemented in this module includes multiple branches. The left branch contains a $1 \times 1$ convolution and adds a combination of normalization and activation functions. The second left branch contains a $1 \times 1$ convolution and adds normalization and activation functions. The combination operation of activation function. The second right branch contains a $1 \times 1$ convolution and the combination operation of normalization and activation function and two $3 \times 3$ convolutions and the combination operation of normalization and activation function. The

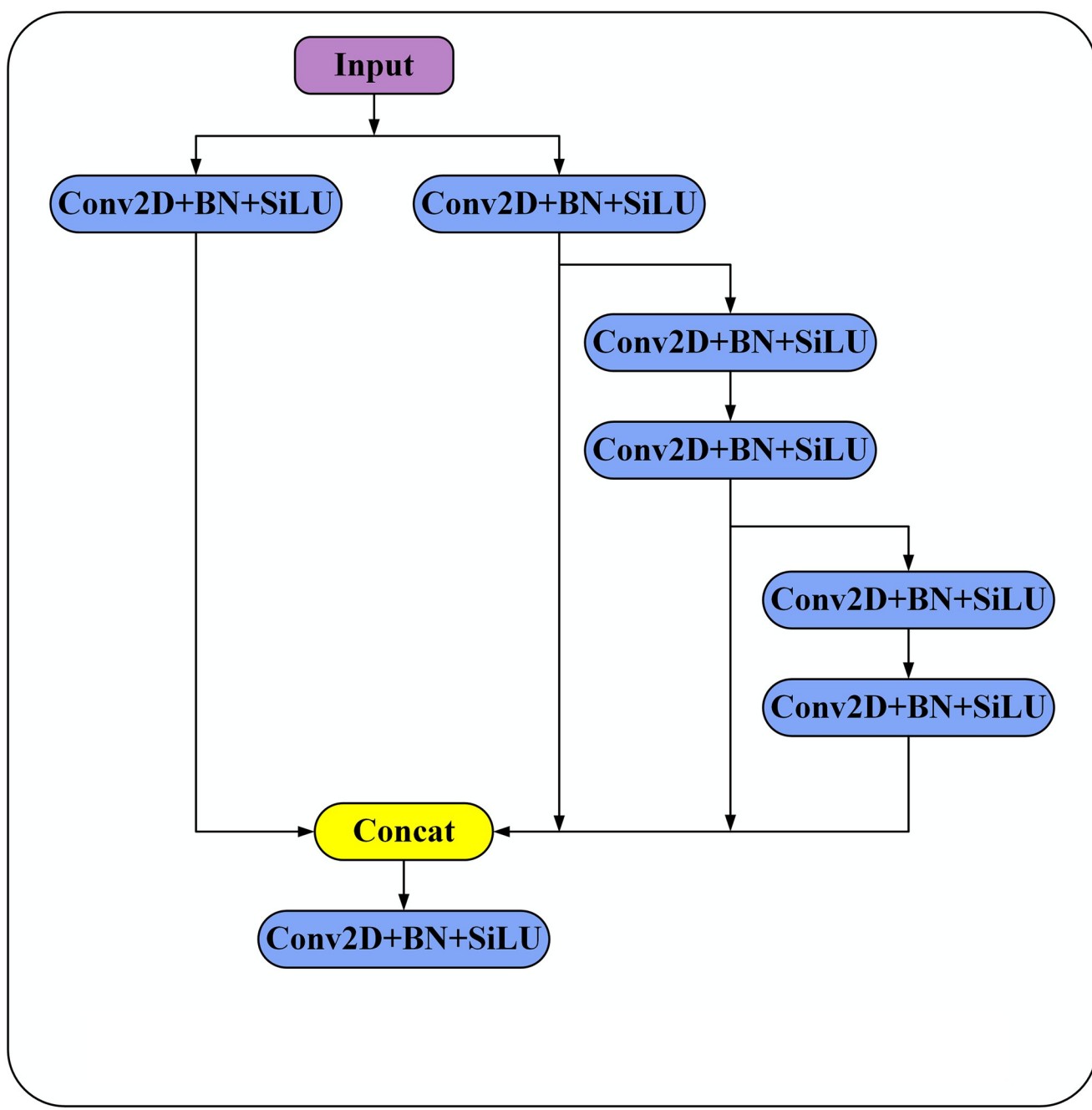

**Fig 4. Multi-branch stacking module structure diagram.**

right branch contains a 1 × 1 convolution and the combination operation of normalization and activation function, two 3 × 3 convolutions and the combination operation of normalization and activation function, and two 5 × 5 convolutions and the combination operation of normalization and activation function, and finally completed After the stacking operation, a 1 × 1 convolution is performed and a combination of normalization and activation functions is added. Convolution operations of different sizes in series and parallel can obtain feature layers with receptive fields of different sizes, which can extract features from target samples of

different sizes, effectively improve the expression of features, and avoid losing relevant features due to convolution operations of uniform size. At the same time, It also helps the algorithm identify and classify different samples to be tested. And Fig 4 is Multi-branch stacking module structure diagram.

## Sampling transition module

In order to promote the communication of feature information between different feature layers, to compensate for the loss of information caused by convolution, and to realize the purpose of cross-channel fusion of feature information. In this paper, a new sampling transition module is designed with the structure shown in Fig 5. When the feature layer is fed into the sampling transition module, it is processed in parallel in two branches. The upper branch consists of a down-sampling module and a $3 \times 3$ convolution with a combination of normalization and activation functions, while in the down-sampling module this paper uses ordinary convolution, maximum pooling, and inverse subpixel convolution to achieve down-sampling. The gain effect of these three operations on feature extraction will be quantitatively demonstrated in the experimental section of Chapter IV. The down branch consists of two $3 \times 3$ convolutions with the combination of normalization and activation functions, and finally the feature information generated by the two branches is stacked. The purpose of using the sampling transition module is to enhance the acquired feature layer by sampling the large-scale feature layer and fusing it with the small-size feature map, while completing the information exchange between different feature layers. At the same time, the sampling transition module will also retain more feature information for the subsequent detection process, avoiding the phenomenon of

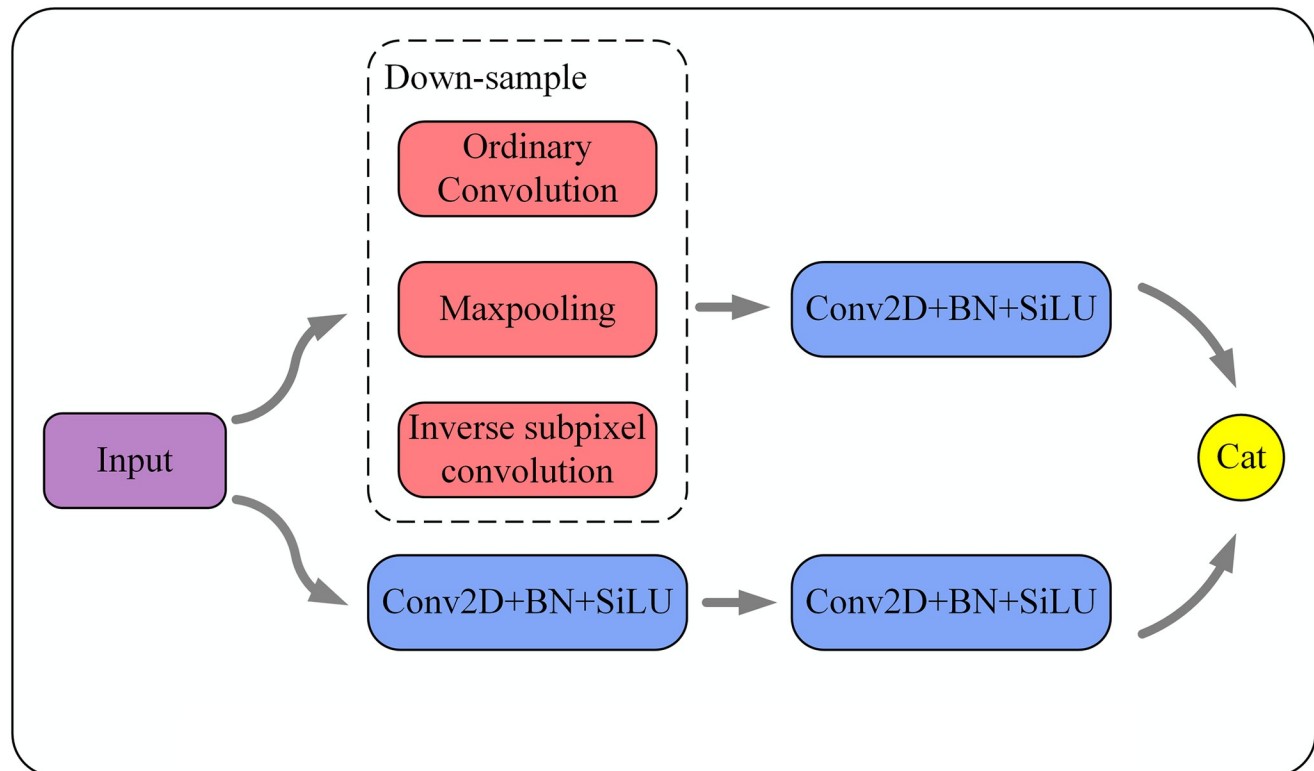

**Fig 5. Sampling transition module structure diagram.**

information loss in the rough down-sampling process. And Fig 5 is Sampling transition module structure diagram.

## Experiments and analysis of results

### Experimental setup

In order to verify the effectiveness of the algorithm AMT-SSD proposed in this article, the PASCAL VOC data set and MS COCO data set are used to verify it in this section. Both the PASCAL VOC data set and the MS COCO data set are universal data sets for verifying target detection algorithms. They contain a total of about 180,000 images of large, medium and small targets, which can reasonably prove whether the algorithm is effective.

During the experimental verification, the experimental platform used was NVIDIA Tesla V100S-PCIE-32GB GPU. Use the Python3.7.6 version compiler to compile the program, and write modular code under the deep learning framework Tensorflow2.2. During the model inference training process, the SGD (Stochastic gradient descent) optimizer is used to help the model accelerate convergence and prevent over-fitting. The specific parameters are listed in Table 1.

In this article, we use two indicators, detection accuracy mAP (mean Average Precision) and detection speed FPS (Frames Per Second), to evaluate the performance of the algorithm model. mAP can intuitively represent the average AP of all categories in the data set, and can avoid the impact of extreme values in certain categories on the evaluation of the overall performance of the algorithm. FPS represents the number of images processed in one second, which can intuitively demonstrate the performance indicator of the detection speed of the algorithm.

The AMT-SSD algorithm is used to draw the loss curve based on the data generated by the loss function during the inference process, which can more intuitively see the convergence process of the training. Fig 6 depicts the training process of the algorithm on the VOC data set. The loss value starts to decrease from around 13.4, and converges around 2 at the end of training. Fig 7 depicts the training process of the algorithm on the MS COCO data set. The loss value starts to decrease from around 10.5, and converges around 3.5 at the end of training. And Fig 6 is AMT-SSD model training loss on VOC dataset, Fig 7 is AMT-SSD model training loss on COCO dataset.

### Analysis of results

In this section, we mainly compare the AMT-SSD algorithm proposed in this article with other target detection algorithms from the experimental quantitative data level. Judging from the experimental results, on the premise of satisfying real-time detection, the AMT-SSD algorithm proposed in this article has good results in detecting large, medium and small targets.

**Experimental results of VOC.**  Table 2 lists the detection accuracy comparison of target detection algorithms based on convolutional neural networks on VOC data sets in recent years. The detection results are all tested on the same dataset.

Table 1. Experimental parameter settings.

| Parameter Name | Parameter Value |
| --- | --- |
| Epoch | 100 |
| Batch size | 16 |
| Patch size | 48 |
| Learning rate | 0.001 |

## A Loss Curve

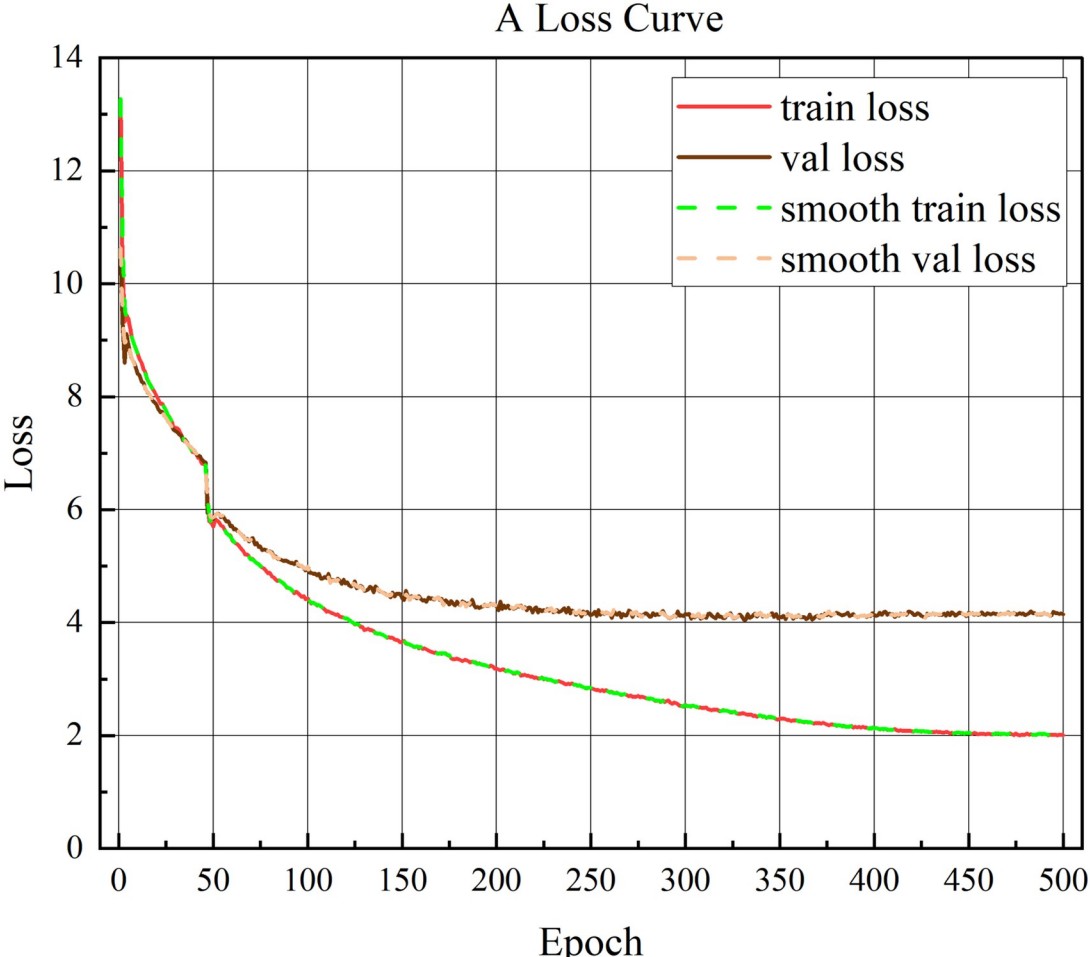

**Fig 6. AMT-SSD model training loss on VOC dataset.**

According to the results in Table 2, it can be concluded that when compared with different algorithms, the AMT-SSD algorithm has improved to varying degrees while meeting the real-time detection requirements. When the input size of the detection image is unified at $300 \times 300$, the detection accuracy of AMT-SSD is 82.4%. Compared with a series of SSD-based algorithms such as SSD, DSSD, MDSSD, DF-SSD, SEFN, RSSD, FSSD, RFB, etc., the detection accuracy is increased by 5.2%,3.8%,3.8%,3.5%,2.8%,3.9%,3.6% and 1.9%, respectively. When the input size of the detection image is unified at $512 \times 512$, the detection accuracy of AMT-SSD is 84.6%. Compared with a series of SSD-based algorithms such as SSD, DSSD, MDSSD, DF-SSD, SEFN, RSSD, FSSD, RFB, ESSD, etc., the detection accuracy is increased by 6.1%,3.1%,3.6%,3.4%,3.8%,3.7%,2.4% and 2.5%, respectively. However, the AMT-SSD algorithm has just reached the threshold of real-time performance in terms of detection speed, and further research is needed.

In order to further illustrate the effectiveness of the AMT-SSD algorithm, in this part, the effectiveness of each module proposed in this article is tested and an ablation experiment is performed on the VOC07+12 data set. Taking the SSD algorithm as the main structure, the Composite Attention Module (CAM), Multi-branch stacking module (MSB) and Sample

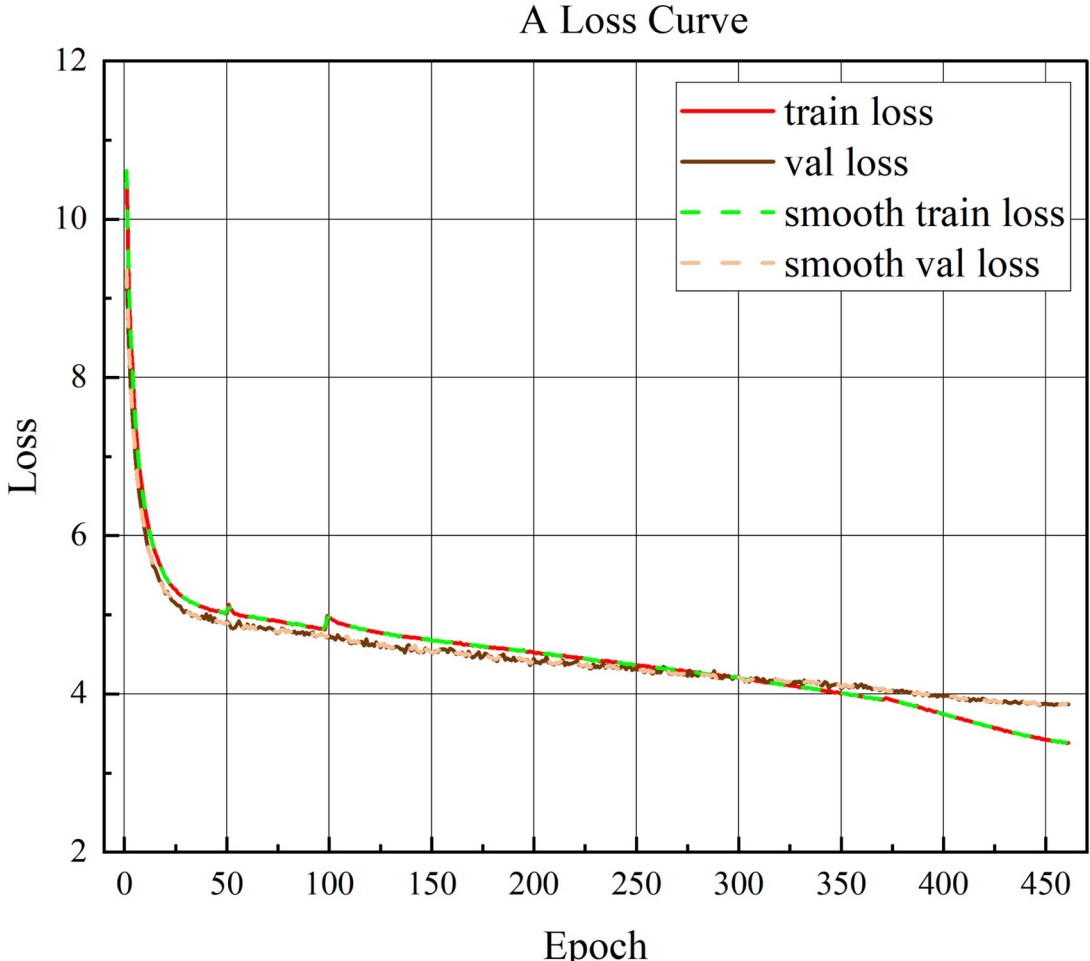

**Fig 7. AMT-SSD model training loss on COCO dataset.**

Transition Block (STB) are added one by one. Through experiments, Quantitative indicators to analyze the effectiveness of each module. The results of the ablation experiment are shown in Table 3.

In order to prove whether CAM is effective, based on the SSD algorithm, a CAM module is added after each extracted feature layer. The input feature information is filtered through CAM to avoid redundant information and background interference from interfering with the algorithm performance. It can be seen from the experimental data that the mAP has increased by 1.7%, which shows that CAM can effectively filter out interference and improve detection accuracy.

From the experimental results, it can be seen that the CAM proposed in this paper is effective. In this regard, we believe that it is the ability of this structure to attend to feature channels and spatial information in a self-learning manner. Useless information can be removed by this structure and interference is minimized to improve the efficiency of the algorithm, which is in line with our original idea. We believe that the target detection task in this paper does not involve stereo targets too much, so we focus more on the sensitivity of the algorithm to channel information. This is the reason for the relatively high number of operations regarding channel

**Table 2. Quantitative comparison of different algorithms on the VOC dataset.**

| Methods | Backbone | Size | mAP(%) | FPS |
|---|---|---|---|---|
| Faster R-CNN [6] | VGG16 | 1000×600 | 73.2 | 7.0 |
| Faster R-CNN [6] | ResNet101 | 1000×600 | 76.4 | 2.4 |
| SSD300 [7] | VGG16 | 300×300 | 77.2 | 46.0 |
| SSD512 [7] | VGG16 | 512×512 | 78.5 | 19.0 |
| DSSD321 [33] | ResNet-101 | 321×321 | 78.6 | 9.5 |
| DSSD513 [33] | ResNet-101 | 513×513 | 81.5 | 5.5 |
| MDSSD300 [34] | VGG16 | 300×300 | 78.6 | 32.2 |
| MDSSD512 [34] | VGG16 | 512×512 | 81.0 | 14.5 |
| DF-SSD [35] | DenseNet-S-32–1 | 300×300 | 78.9 | 11.6 |
| SEFN300 [36] | VGG16 | 300×300 | 79.6 | 55.0 |
| SEFN512 [36] | VGG16 | 512×512 | 81.2 | 30.0 |
| RSSD300 [37] | VGG16 | 300×300 | 78.5 | 35.0 |
| RSSD512 [37] | VGG16 | 512×512 | 80.8 | 16.6 |
| FSSD300 [38] | VGG16 | 300×300 | 78.8 | 65.8 |
| FSSD512 [38] | VGG16 | 512×512 | 80.9 | 35.7 |
| RFB300 [39] | VGG16 | 300×300 | 80.5 | 83.0 |
| RFB512 [39] | VGG16 | 512×512 | 82.2 | 38.0 |
| ESSD [40] | VGG16 | 512×512 | 82.1 | 15.7 |
| AMT-SSD300 | VGG16 | 300×300 | **82.4** | 26.1 |
| AMT-SSD512 | VGG16 | 512×512 | **84.6** | 17.4 |

feature extraction in the CAM structure. In this structure, the strength of the beneficial features in the channel is evaluated and from this aspect, the network model is judged whether the features need to be suppressed or boosted. And the presence of pooling operations in the structure, either global average pooling or global maximum pooling, uninterruptedly extracts for the algorithm the features with the most drastic expression within a certain range; and at the same time, the scale of the feature map is reduced, which captures a large range of feature information for the algorithm. The purpose of focusing on spatial information is to enhance the network's representation of the characteristics of different spatial locations, utilizing the spatial relationships between different pixel points so that the network can better focus on spatial locations that are beneficial to the task at hand. The CAM proposed in this paper can do an effective balance between channel and spatial information, which makes the CAM perform well.

In order to prove whether MSB is effective, we added MSB to the first three shallow feature layers of CAM to capture its rich details and spatial information, and further utilize the features information of small targets through the multi-branch structure in MSB. After adding

**Table 3. Analysis of ablation experiment results.**

| CAM | MSB | STB | VOC 07+12 | | |
|---|---|---|---|---|---|
| | | | mAP/% | | |
| | | | 72.4 | | |
| ✓ | | | 74.1 | | |
| ✓ | ✓ | | 77.5 | | |
| ✓ | ✓ | ✓ | 82.4 | | |

MSB, the mAP value increased by 3.4%, which shows that MSB has a good effect in capturing target feature information and enhancing its expression effect.

The effectiveness of MSB we believe is contributed by the multiple branches of the structure, each of which has a convolutional layer with a different convolutional kernel to extract features. If a single branch is used to perform extraction operations on features, this inevitably produces feature loss, which is naturally compensated by the structure of the MSB. Each branch has a different convolutional kernel, which makes the features extracted from each branch different and effective. The operation of stacking feature layers in MSB is to provide a guarantee that the algorithm does not lose feature information.

In order to prove whether STB is effective, after adding CAM and MSB at the corresponding positions, we also added STB before the final detection output of each feature layer, and the mAP increased by 4.9%. This shows that STB plays a positive role in promoting feature communication and making up for lost feature information during the down-sampling process.

At the same time, in order to verify the effectiveness of the three down-sampling methods mentioned in the new sampling transition module for feature extraction, relevant comparative experiments were conducted. While keeping the positions and quantities added by each module unchanged, only the down-sampling method in the oversampling module is changed. The results are shown in Table 4. When using maximum pooling for down–sampling, the lowest mAP of the overall algorithm is only 78.1%. When using ordinary convolution for down-sampling, the mAP of the algorithm reaches 81.2%, which is better than maximum pooling in terms of feature expression. When using inverse sub-pixel convolution to down-sample the feature map, the mAP reached 82.4%. Compared with max pooling and ordinary convolution, the performance is improved by 4.3% and 1.2% respectively. Judging from the results, down-sampling the feature map by using inverse sub-pixel convolution has a better gain effect on the expression of features.

From the experimental results in Table 4, the inverse subpixel convolution has good feature extraction. We believe that this is due to its utilization of its own feature information to change the dimension of the feature map. Inverse subpixel convolution can reduce the size of the feature map while effectively utilizing the feature information to achieve feature scale balance. The specific process of inverse sub-pixel convolution is shown in Fig 8. Unlike ordinary convolution and maximum pooling, which will lose feature information when down-sampling, inverse sub-pixel convolution will rearrange feature information that does not meet the output size according to certain rules when down-sampling. While down-sampling is implemented, the integrity of feature information is also ensured. And Fig 8 is diagram of Inverse sub-pixel convolution process.

**Qualitative experiments comparison.** In this section, the good effect of this algorithm in the actual detection process is mainly reflected from the perspective of qualitative experiments. In Fig 9, the AMT-SSD algorithm performs comparative experiments with the SSD algorithm, SEFN algorithm, and ESSD algorithm respectively. From the results, it can be seen that the detection effect of the SSD algorithm is obviously insufficient compared with the above-

**Table 4. Comparison of results for different sampling methods.**

| Down-sample Modul | Map/% |
|---|---|
| Conv | 81.2 |
| Max-pooling | 78.1 |
| Inverse subpixel convolution | 82.4 |

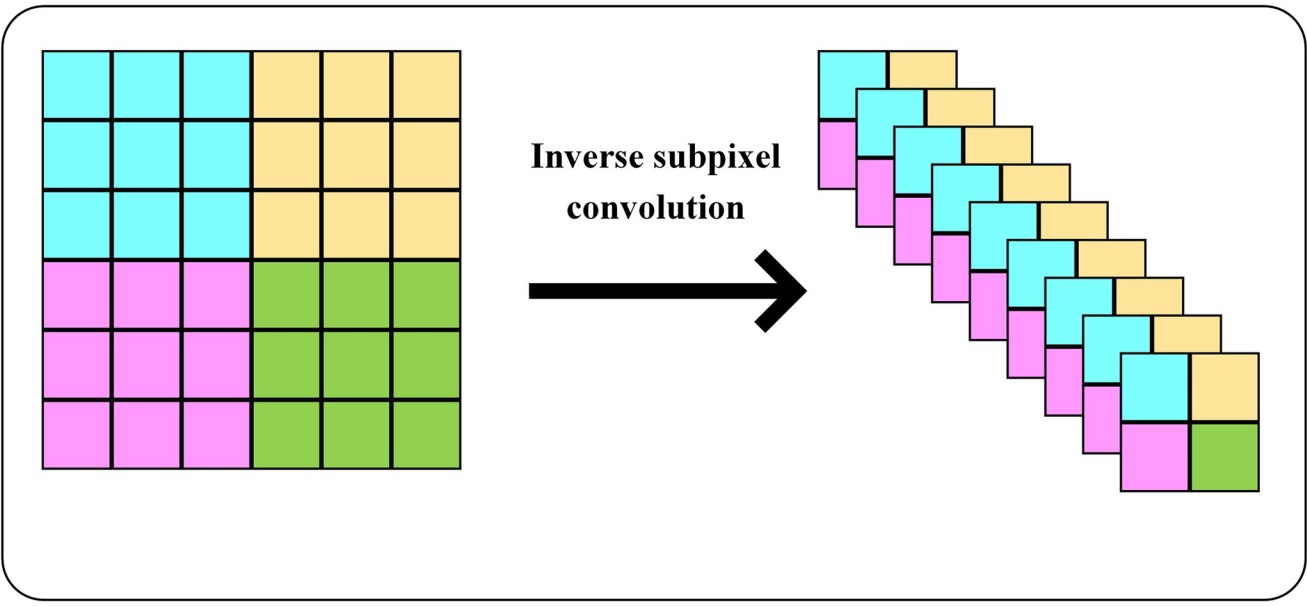

**Fig 8. The diagram of inverse sub-pixel convolution process.**

mentioned algorithms, and there is a serious missed detection phenomenon, although the SEFN algorithm and ESSD algorithm have alleviated this phenomenon to a certain extent, the overall detection accuracy is still slightly insufficient. The AMT-SSD algorithm can not only solve the problem of missed detection, but also achieve satisfactory results in detection accuracy. From the data indicators in the previous section and the qualitative experiments in this section, we can see that the design of the AMT-SSD algorithm and the arrangement of functional modules are reasonable and effective. And Fig 9 is diagram of Qualitative experimental comparison of different algorithms.

**Experimental results of COCO.** In order to better prove that the AMT-SSD algorithm has good detection performance in the target detection process, in this section, the MS COCO data set is used as another comparative experiment and the obtained quantitative data are compared and the results are shown in Table 5. It can be seen from the data listed in Table 5 that compared with other algorithms, both the detection accuracy and recall rate of the AMT-SSD algorithm have been improved to a certain extent, which also shows that the AMT-SSD algorithm has good performance in target detection.

## Conclusion

A small target detection algorithm AMT-SSD that combines the multi-branch stacking of the attention mechanism and the new sampling transition module proposed in this paper. And the AMT-SSD improves the detection performance of small targets while improving the target feature expression and reducing the loss of feature information during the sampling process. The whole system consists of three parts: composite attention machine module, multi-branch stacking module and sampling transition module. First, the composite attention module is used to filter useless information and improve detection efficiency and algorithm robustness by improving correlation in both space and channel; In the multi-branch stacking module, the parallel branch structure is used to fuse convolution kernels of different sizes to fully extract

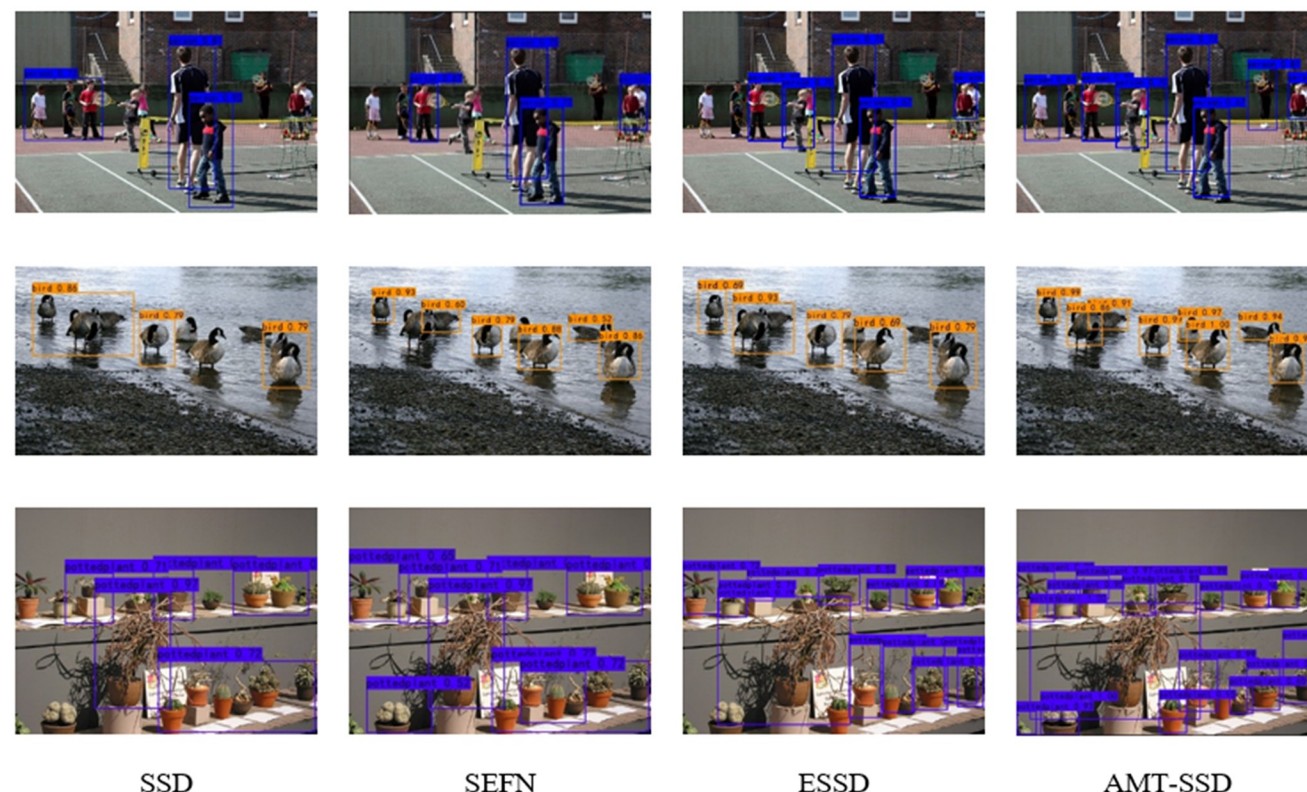

**Fig 9. The diagram of Qualitative experimental comparison of different algorithms.**

target features and enhance the feature expression effect; In addition, a new sampling transition module that incorporates inverse sub-pixel convolution is used during sampling to compensate for the loss of features during the generation of feature maps. Through experimental analysis, on the PASCAL VOC and MS COCO data sets, the AMT-SSD algorithm proposed in this article achieved mAP of 84.6% and 53.4% respectively, and achieved an FPS of 26.1. The results show that the AMT-SSD proposed in this article has good detection performance in small target detection. However, the research in this paper does not make strict requirements on the detection speed of the algorithm, and the algorithm in this paper only achieves the

**Table 5. Analysis of ablation experiment results.**

| Methods | Backbone | Avg. precision, IoU | | | Avg. precision, area | | | Avg. recall, area | | |
|---|---|---|---|---|---|---|---|---|---|---|
| | | IOU = 0.5:0.95 | IOU = 0.5 | IOU = 0.75 | Area:S | Area:M | Area:L | Area:S | Area:M | Area:L |
| SEFN512 [36] | VGG16 | 33.7 | 54.7 | 35.6 | **19.2** | 38.0 | 47.3 | **29.1** | **52.5** | 63.2 |
| SSD512 [18] | VGG16 | 27.7 | 46.4 | 26.7 | 10.9 | 31.8 | 43.5 | 16.5 | 46.6 | 60.8 |
| FSSD512 [38] | VGG16 | 31.8 | 52.8 | 33.5 | 14.2 | 35.1 | 45.0 | 22.3 | 49.9 | 62.0 |
| DF-SSD [35] | DenseNet-S-32–1 | 29.5 | 50.7 | 31.3 | 9.8 | 31.1 | 46.5 | 17.3 | 46.8 | 64.4 |
| DSOD300 [41] | DS/64–192–48–1 | 29.3 | 47.3 | 30.6 | 9.4 | 31.5 | 47.0 | 16.7 | 47.1 | 65.0 |
| DSSD513 [33] | ResNet-101 | 33.2 | 53.3 | 35.2 | 13.0 | 35.4 | 51.1 | 28.9 | 43.5 | 46.2 |
| RFB512 [39] | VGG16 | 34.4 | 55.7 | 36.4 | 17.6 | 37.0 | 47.6 | 27.3 | **52.3** | 65.4 |
| AMT-SSD | VGG16 | **53.4** | **80.9** | **59.6** | **19.6** | **38.7** | **61.7** | **29.7** | 51.3 | **70.3** |

demand of real-time detection. Therefore, how to improve the detection speed of the algorithm and reduce the number of parameters of the algorithm will be our further research.

## Supporting information

**S1 File. The contents of the supporting information file include: Modified images, quantitative data, and experimental metrics.**
(ZIP)

## Author Contributions

**Writing – original draft:** Qingyao Lin.

**Writing – review & editing:** Rugang Wang, Yuanyuan Wang, Feng Zhou.

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
