## [Decision Letter · Decision Letter 0]

18 Jan 2024

PONE-D-23-42468Small target detection algorithm based on multi-branch stacking and new sampling transition modulePLOS ONE

Dear Dr. 林庆耀,

Thank you for submitting your manuscript to PLOS ONE. After careful consideration, we feel that it has merit but does not fully meet PLOS ONE’s publication criteria as it currently stands. Therefore, we invite you to submit a revised version of the manuscript that addresses the points raised during the review process.

We look forward to receiving your revised manuscript.

Kind regards,

Narendra Khatri, Ph.D.

Academic Editor

PLOS ONE

Journal Requirements:

   "This work was supported by the Jiangsu Graduate Practical Innovation Project (No: SJCX22_1685), the Major Project of Natural Science Research of Jiangsu Province Colleges and Universities (No: 19KJA110002), the Natural Science Foundation of China under Grant (No. 61673108), the Natural Science Research Project of Jiangsu University (NO. 18KJD510010)."

5. We note that you have indicated that there are restrictions to data sharing for this study. PLOS only allows data to be available upon request if there are legal or ethical restrictions on sharing data publicly. For more information on unacceptable data access restrictions, please see http://journals.plos.org/plosone/s/data-availability#loc-unacceptable-data-access-restrictions. 

Additional Editor Comments:

Your article would appear to be of interest to a wide engineering research community and in order to promote its visibility even more, may we recommend that you view the past published articles in PLOS ONE and if you find any relevant publications, CITE the article from this Journal.

1. Results and discussion: A critical analysis of the obtained results is missing; authors must analyse the obtained results by comparing it with the similar existing model’s results.

2. The conclusion must be of maximum 200 words and it must present the major findings with the help of the quantitative (numbers) in support of the claims.

Reviewers' comments:

Reviewer's Responses to Questions

**Comments to the Author**

1. Is the manuscript technically sound, and do the data support the conclusions?

Reviewer #1: Yes

2. Has the statistical analysis been performed appropriately and rigorously? 

Reviewer #1: Yes

3. Have the authors made all data underlying the findings in their manuscript fully available?

Reviewer #1: Yes

4. Is the manuscript presented in an intelligible fashion and written in standard English?

Reviewer #1: Yes

5. Review Comments to the Author

Reviewer #1: - My big concern is the “discussion” part. I couldn’t see any discussion. In discussion, what I expect to see is that what are the plausible reasons behind obtaining these results? Moreover, the validity and accuracy of the data should be discussed. The authors must comment on whether or not the results were expected and present explanations for the results; they must go into greater depth when explaining findings that were unexpected or especially profound. If appropriate, note any unusual or unanticipated patterns or trends that emerged from your results and explain their meaning. Be sure to advocate for your findings and underline how your results significantly in move the field forward. Remember to make sure you give your results their due and not undermine them. Moreover, in discussion, you should clearly state what your study adds to the body of the literature. In discussion, the authors must explain why these results were obtained? What is/are the plausible reasons behind obtaining these results? In addition, in discussion, the authors must contrast the results of your study with those have been previously done.

- Conclusion is not enough. It is not to the point. The conclusion must help the reader understand why your research should matter to them after they have finished reading the paper. A conclusion is not merely a summary of your points or a re-statement of your research problem but a synthesis of key points.

6. PLOS authors have the option to publish the peer review history of their article (what does this mean?). If published, this will include your full peer review and any attached files.

Reviewer #1: **Yes: **Prathamesh P Churi

---

## [Author Response · Author response to Decision Letter 0]

30 Jan 2024

Dear Editor:

Re: Manuscript (PONE-D-23-42468) – “Small target detection algorithm based on multi-branch stacking and new sampling transition module.”, by Qingyao Lin, Rugang Wang, Yuanyuan Wang, Feng Zhou. 

Thank you very much for your kind consideration of our manuscript. We have carefully studied the comments raised by the reviewers and have made amendments in accordance with the comments and suggestions from the reviewers. All the changes made in the revised version are underlined. Our responses to the reviewers’ comments and suggestions are enclosed in this letter. 

We look forward to hearing a favorable reply from you. 

Yours sincerely, 

Qingyao Lin

 

Reply to Reviewer 1

Dear Reviewer,

We would like to express our sincere thanks to the reviewer for his/her valuable comments and suggestions. We have revised the manuscript in accordance with the reviewer’s comments and suggestions. All the changes made in the revision are underlined. Our replies to the reviewer’s comments and suggestions are as follows.

Comment #1-1:

My big concern is the “discussion” part. I couldn’t see any discussion. In discussion, what I expect to see is that what are the plausible reasons behind obtaining these results? Moreover, the validity and accuracy of the data should be discussed. The authors must comment on whether or not the results were expected and present explanations for the results; they must go into greater depth when explaining findings that were unexpected or especially profound. If appropriate, note any unusual or unanticipated patterns or trends that emerged from your results and explain their meaning. Be sure to advocate for your findings and underline how your results significantly in move the field forward. Remember to make sure you give your results their due and not undermine them. Moreover, in discussion, you should clearly state what your study adds to the body of the literature. In discussion, the authors must explain why these results were obtained? What is/are the plausible reasons behind obtaining these results? In addition, in discussion, the authors must contrast the results of your study with those have been previously done.

Conclusion is not enough. It is not to the point. The conclusion must help the reader understand why your research should matter to them after they have finished reading the paper. A conclusion is not merely a summary of your points or a re-statement of your research problem but a synthesis of key points.

Response #1-1:

The reviewer’s comment is appreciated. 

We agree with the reviewer's view that the discussion section is missing. It will be supplemented by an explanation of the role of the CAM module and the MSB and the reasons for this in the ablation experiment section in chapter 4.2. The discussion of the STB module and why we chose inverse subpixel convolution is placed below Table 4, which we will highlight in red in the original text. The specifics of the additions we made are as follows and will be highlighted in red in the manuscript.

From the experimental results, it can be seen that the CAM proposed in this paper is effective. In this regard, we believe that it is the ability of this structure to attend to feature channels and spatial information in a self-learning manner. Useless information can be removed by this structure and interference is minimized to improve the efficiency of the algorithm, which is in line with our original idea. We believe that the target detection task in this paper does not involve stereo targets too much, so we focus more on the sensitivity of the algorithm to channel information. This is the reason for the relatively high number of operations regarding channel feature extraction in the CAM structure. In this structure, by detecting beneficial features in the channel, which is the strength of the current feature response that is weighted and evaluated. This weight is utilized to judge the importance of features and suppress or enhance them. And the presence of pooling operations in the structure, either global average pooling or global maximum pooling, uninterruptedly extracts for the algorithm the features with the most drastic expression within a certain range; and at the same time, the scale of the feature map is reduced, which captures a large range of feature information for the algorithm. The purpose of focusing on spatial information is to enhance the network's representation of the characteristics of different spatial locations, utilizing the spatial relationships between different pixel points so that the network can better focus on spatial locations that are beneficial to the task at hand. The CAM proposed in this paper can do an effective balance between channel and spatial information, which makes the CAM perform well.

The effectiveness of MSB we believe is contributed by the multiple branches of the structure, each of which has a convolutional layer with a different convolutional kernel to extract features. If a single branch is used to perform extraction operations on features, this inevitably produces feature loss, which is naturally compensated by the structure of the MSB. Each branch has a different convolutional kernel, which makes the features extracted from each branch different and effective. The operation of stacking feature layers in MSB is to provide a guarantee that the algorithm does not lose feature information.

---

## [Decision Letter · Decision Letter 1]

12 Apr 2024

PONE-D-23-42468R1Small target detection algorithm based on multi-branch stacking and new sampling transition modulePLOS ONE

Dear Dr. 林庆耀,

Thank you for submitting your manuscript to PLOS ONE. After careful consideration, we feel that it has merit but does not fully meet PLOS ONE’s publication criteria as it currently stands. Therefore, we invite you to submit a revised version of the manuscript that addresses the points raised during the review process.

**ACADEMIC EDITOR: **The paper still requires corrections, as several points raised by the reviewers need to be addressed before it can be further considered.

We look forward to receiving your revised manuscript.

Kind regards,

Narendra Khatri, Ph.D.

Academic Editor

PLOS ONE

Reviewers' comments:

Reviewer's Responses to Questions

**Comments to the Author**

1. If the authors have adequately addressed your comments raised in a previous round of review and you feel that this manuscript is now acceptable for publication, you may indicate that here to bypass the “Comments to the Author” section, enter your conflict of interest statement in the “Confidential to Editor” section, and submit your "Accept" recommendation.

Reviewer #2: (No Response)

Reviewer #3: All comments have been addressed

2. Is the manuscript technically sound, and do the data support the conclusions?

Reviewer #2: (No Response)

Reviewer #3: Partly

3. Has the statistical analysis been performed appropriately and rigorously? 

Reviewer #2: (No Response)

Reviewer #3: I Don't Know

4. Have the authors made all data underlying the findings in their manuscript fully available?

Reviewer #2: (No Response)

Reviewer #3: Yes

5. Is the manuscript presented in an intelligible fashion and written in standard English?

Reviewer #2: (No Response)

Reviewer #3: No

6. Review Comments to the Author

Reviewer #2: 1. The motivations of the paper are not clear. The disadvantages of the existing schemes must be mentioned.

2. Contributions are not mentioned. Most importantly, the structure of the Introduction section is very poor.

3. Related schemes are not discussed properly. The following papers must be included:

Performance releaser with smart anchor learning for arbitrary-oriented object detection

Machine learning insights into hypersonics research evolution: A 21st century perspective

IFODPSO-based multi-level image segmentation scheme aided with Masi entropy

Needle detection and localisation for robot-assisted subretinal injection using deep learning

Data accessing based on the popularity value for cloud computing

Leveraging deep learning techniques to obtain efficacious segmentation results

QEST: Quantized and efficient scene text detector using deep learning

4. Algorithm or step-wise discussion of the proposed scheme is not mentioned.

5. The proposed scheme is unstructured. It is hard to identify the novelty of the proposed work.

6. Equations and figures are not represented properly. Also, the key terms of many equations are not defined.

7. Technical discussion on results is not given. Moreover, the results are not convincing.

8. The English language is very poor.

9. The organization of the paper is poor.

10. Important references are missing and all the details of the references are not given.

11. Delete Author Summary

Reviewer #3: 1.) I couldn't find the chapter 4.2 as highlighted by the author in their comments, " the ablation experiment section in chapter 4.2.".

2) There are many vague and irrelevant sentences which do not convey any significant meaning: For example, "In this structure, by detecting beneficial features in the channel, which is the strength of the current feature response that is weighted and evaluated..."

3)The structure and sentence formation is too long which is breaking the true meaning that author is trying to explain. Multiple things are explained in long statements which diverts the reader to understand properly.

4) The references are written in haphazard manner not following proper coherence or uniformity.

5) The first line of the Conclusion section : In order to solve the problem of poor feature expression and easy loss of feature information during the sampling process, resulting in poor performance in small target detection. " seems to be incomplete and over lengthy breaking the exact meaning author wants to explain.

6) The way this paragraph is written: We believe that this is because compared with the down-sampling methods of

maximum pooling and ordinary convolution, inverse sub-pixel convolution uses the

feature information of the feature map itself to directly change the scale, rather than

through convolution operations and take the maximum value to fill in the feature

information" sounds very casual and informal. Again diverts the reader from exact information

7. PLOS authors have the option to publish the peer review history of their article (what does this mean?). If published, this will include your full peer review and any attached files.

Reviewer #2: No

Reviewer #3: No

---

## [Author Response · Author response to Decision Letter 1]

16 May 2024

Dear Editor:

Re: Manuscript (PONE-D-23-42468R1) – “Small target detection algorithm based on multi-branch stacking and new sampling transition module.”, by Qingyao Lin, Rugang Wang, Yuanyuan Wang, Feng Zhou. 

Thank you very much for your kind consideration of our manuscript. We have carefully studied the comments raised by the reviewers and have made amendments in accordance with the comments and suggestions from the reviewers. All changes in the revised version will be highlighted in red. Our responses to the reviewers’ comments and suggestions are enclosed in this letter. 

We look forward to hearing a favorable reply from you. 

Yours sincerely, 

Qingyao Lin

 

Reply to Reviewer 2

Dear Reviewer,

We would like to express our sincere thanks to the reviewer for his/her valuable comments and suggestions. We have revised the manuscript in accordance with the reviewer’s comments and suggestions. All the changes made in the revision are underlined. Our replies to the reviewer’s comments and suggestions are as follows.

Comment #2-1:

The motivations of the paper are not clear. The disadvantages of the existing schemes must be mentioned.

Response #2-1:

The reviewer’s comment is appreciated. The motivation of this paper is to address the problem that the SSD algorithm does not adequately extract the feature information contained in each feature layer and that information is lost during the sampling process. We describe these elements in the abstract section. The disadvantage of the research in this paper is that the detection speed of the algorithm is not strictly required, thus making the algorithm only initially capable of real-time detection. We will add to the above shortcoming of the algorithm in the CONCLUSION section.

Comment #2-2:

Contributions are not mentioned. Most importantly, the structure of the Introduction section is very poor.

Response #2-2:

The reviewer’s comment is appreciated. The contribution is mainly described in the third paragraph of the introduction section. Perhaps our presentation was not clear, so we have modified this section to highlight the content of the contribution.

Comment #2-3:

Related schemes are not discussed properly. The following papers must be included:

Performance releaser with smart anchor learning for arbitrary-oriented object detection

Machine learning insights into hypersonics research evolution: A 21st century perspective

IFODPSO-based multi-level image segmentation scheme aided with Masi entropy

Needle detection and localisation for robot-assisted subretinal injection using deep learning

Data accessing based on the popularity value for cloud computing

Leveraging deep learning techniques to obtain efficacious segmentation results

QEST: Quantized and efficient scene text detector using deep learning

Response #2-3:

The reviewer’s comment is appreciated. We have discussed the above article and placed it in the INTRODUCTION section. The details are as follows:

There are many important downstream branch tasks in the field of computer vision, and target detection, as an important one, has achieved remarkable results in various fields. Zhang[1]et al. proposed an ALS-Performance Releaser (PRSAL) with the learning function of intelligent anchors for anchor frame learning strategies. It utilizes anchor classification ability as an equivalent indicator of anchor box regression ability to screen anchors with high detection potential in a more rational way. Chakraborty R[2]et al. proposed an improved version of Fractional Order Darwinian PSO (IFODPSO) for segmenting histogram based 3D color images on multiple levels of Berkeley Segmentation Data Set (BSDS500). This scheme overcomes the complete dependence on score coefficients when dealing with multilevel problems with datasets, which provides new ideas for studying computer vision segmentation tasks. Zhou M[3]et al. demonstrated a robust framework for needle detection and localization in robot-assisted subretinal injections using a cross-cutting study between deep learning and medical contexts. The best performing network successfully detected and localized all needles in the dataset with an IoU value of 0.55 when evaluated on live pig eyes. Manjari K[4]et al. developed a novel lightweighting model with ResNet50 and MobileNetV2 as the backbone, which improves the efficiency of scene text detection and reduces the resource cost. It provides ideas for our subsequent research on network lightweighting.

Comment #2-4:

Algorithm or step-wise discussion of the proposed scheme is not mentioned.

Response #2-4:

The reviewer’s comment is appreciated. The proposed scheme is based on the SSD algorithm. And the step-wise discussion we have described in the ablation experiment section.

Comment #2-5:

The proposed scheme is unstructured. It is hard to identify the novelty of the proposed work.

Response #2-5:

The reviewer’s comment is appreciated. The structural diagram of the proposed scheme we give in Fig. 2. The implementation and functionality of each module is related in Analysis of the AMT-SSD algorithm model section. Among them, MC Block and Tran Block are new modules with innovative design in SSD algorithm.

Comment #2-6:

Equations and figures are not represented properly. Also, the key terms of many equations are not defined.

Response #2-6:

The reviewer’s comment is appreciated. The formulas and numerical expressions that appear in the Composite Attention Module section represent only the implementation of the CAM module. Where equations and figures are missing from the text, we will complete them.

Comment #2-7:

Technical discussion on results is not given. Moreover, the results are not convincing.

Response #2-7:

The reviewer’s comment is appreciated. The discussion of the results we describe in the ablation experiments section. In this section we provide a detailed description of the role of each module and the reasons for the advantages achieved.

Comment #2-8:

The English language is very poor.

Response #2-8:

The reviewer’s comment is appreciated. We have embellished and revised the language.

Comment #2-9:

The organization of the paper is poor.

Response #2-9:

The reviewer’s comment is appreciated. We revised the language of the paper as well as the relevant statements and did our best to make the paper logical and well-organized.

Comment #2-10:

Important references are missing and all the details of the references are not given.

Response #2-10:

The reviewer’s comment is appreciated. Missing relevant literature we will add and elaborate in detail in the introduction section. These will be revised in tandem with Comment #2-3.

Comment #2-11:

Delete Author Summary.

Response #2-11:

The reviewer’s comment is appreciated. The Author Summary are deleted.

Reply to Reviewer 3

Dear Reviewer,

We would like to express our sincere thanks to the reviewer for his/her valuable comments and suggestions. We have revised the manuscript in accordance with the reviewer’s comments and suggestions. All the changes made in the revision are underlined. Our replies to the reviewer’s comments and suggestions are as follows.

Comment #3-1:

I couldn't find the chapter 4.2 as highlighted by the author in their comments, " the ablation experiment section in chapter 4.2.".

Response #3-1:

The reviewer’s comment is appreciated. I'm very sorry for the inconvenience. The contents of the ablation experiments are shown in Table 3, and the related analyses are also presented near Table 3.

Comment #3-2:

There are many vague and irrelevant sentences which do not convey any significant meaning: For example, "In this structure, by detecting beneficial features in the channel, which is the strength of the current feature response that is weighted and evaluated..."

Response #3-2:

The reviewer’s comment is appreciated. We have revised the relevant statements to ensure that a clearer meaning can be conveyed. The details are as follows:

In this structure, the strength of the beneficial features in the channel is evaluated and from this aspect, the network model is judged whether the features need to be suppressed or boosted.

Comment #3-3:

The structure and sentence formation is too long which is breaking the true meaning that author is trying to explain. Multiple things are explained in long statements which diverts the reader to understand properly.

Response #3-3:

The reviewer’s comment is appreciated. We have revised excessively long statements in the text and reorganized the narrative for simplicity and clarity.

Comment #3-4:

The references are written in haphazard manner not following proper coherence or uniformity.

Response #3-4:

The reviewer’s comment is appreciated. We would like to cover as much as possible of the research program and research background in terms of references. So the references covered mainly the development of SSD algorithms for target detection. It also contains some research content for small targets, although these research programs do not use SSD algorithms, but provide ideas for small target research.

Comment #3-5:

The first line of the Conclusion section: In order to solve the problem of poor feature expression and easy loss of feature information during the sampling process, resulting in poor performance in small target detection. " seems to be incomplete and over lengthy breaking the exact meaning author wants to explain.

Response #3-5:

The reviewer’s comment is appreciated. In response to this problem, we have made changes to reorganize and recount the statements. The details are as follows:

A small target detection algorithm AMT-SSD that combines the multi-branch stacking of the attention mechanism and the new sampling transition module proposed in this paper. And the AMT-SSD improves the detection performance of small targets while improving the target feature expression and reducing the loss of feature information during the sampling process.

Comment #3-6:

The way this paragraph is written: We believe that this is because compared with the down-sampling methods of maximum pooling and ordinary convolution, inverse sub-pixel convolution uses the feature information of the feature map itself to directly change the scale, rather than through convolution operations and take the maximum value to fill in the feature information" sounds very casual and informal. Again diverts the reader from exact information.

Response #3-6:

The reviewer’s comment is appreciated. We have revised the expression to convey the exact meaning to the readers in a more formal and rigorous way.

From the experimental results in Table 4, the inverse subpixel convolution has good feature extraction. We believe that this is due to its utilization of its own feature information to change the dimension of the feature map. Inverse subpixel convolution can reduce the size of the feature map while effectively utilizing the feature information to achieve feature scale balance.

---

## [Decision Letter · Decision Letter 2]

28 May 2024

Small target detection algorithm based on multi-branch stacking and new sampling transition module

PONE-D-23-42468R2

Dear Dr. 林庆耀,

We’re pleased to inform you that your manuscript has been judged scientifically suitable for publication and will be formally accepted for publication once it meets all outstanding technical requirements.

Kind regards,

Narendra Khatri, Ph.D.

Academic Editor

PLOS ONE

Additional Editor Comments (optional):

Accept

Reviewers' comments:

Reviewer's Responses to Questions

**Comments to the Author**

1. If the authors have adequately addressed your comments raised in a previous round of review and you feel that this manuscript is now acceptable for publication, you may indicate that here to bypass the “Comments to the Author” section, enter your conflict of interest statement in the “Confidential to Editor” section, and submit your "Accept" recommendation.

Reviewer #2: All comments have been addressed

2. Is the manuscript technically sound, and do the data support the conclusions?

Reviewer #2: Yes

3. Has the statistical analysis been performed appropriately and rigorously? 

Reviewer #2: Yes

4. Have the authors made all data underlying the findings in their manuscript fully available?

Reviewer #2: Yes

5. Is the manuscript presented in an intelligible fashion and written in standard English?

Reviewer #2: Yes

6. Review Comments to the Author

Reviewer #2: The authors have properly addressed all the comments. Therefore, this paper can be considered for publication.

7. PLOS authors have the option to publish the peer review history of their article (what does this mean?). If published, this will include your full peer review and any attached files.

Reviewer #2: No

---

## [Editor Report · Acceptance letter]

11 Jul 2024

PONE-D-23-42468R2 

PLOS ONE

Dear Dr. Lin, 

I'm pleased to inform you that your manuscript has been deemed suitable for publication in PLOS ONE. Congratulations! Your manuscript is now being handed over to our production team.

Kind regards, 

on behalf of

Dr. Narendra Khatri 

Academic Editor

PLOS ONE